# Forecasting COVID-19 Case Trends Using SARIMA Models during the Third Wave of COVID-19 in Malaysia

**DOI:** 10.3390/ijerph19031504

**Published:** 2022-01-28

**Authors:** Cia Vei Tan, Sarbhan Singh, Chee Herng Lai, Ahmed Syahmi Syafiq Md Zamri, Sarat Chandra Dass, Tahir Bin Aris, Hishamshah Mohd Ibrahim, Balvinder Singh Gill

**Affiliations:** 1Institute for Medical Research (IMR), Ministry of Health Malaysia, Shah Alam 40170, Malaysia; lssarbhan@moh.gov.my (S.S.); jochua0505@gmail.com (C.H.L.); syahmi.syafiq@moh.gov.my (A.S.S.M.Z.); tahir.a@moh.gov.my (T.B.A.); drbsgill@moh.gov.my (B.S.G.); 2School of Mathematical and Computer Sciences, Heriot-Watt University Malaysia, Putrajaya 62200, Malaysia; s.dass@hw.ac.uk; 3Ministry of Health, Malaysia, Putrajaya 62590, Malaysia; drhishamshah@moh.gov.my

**Keywords:** COVID-19, forecast, ARIMA, Malaysia

## Abstract

With many countries experiencing a resurgence in COVID-19 cases, it is important to forecast disease trends to enable effective planning and implementation of control measures. This study aims to develop Seasonal Autoregressive Integrated Moving Average (SARIMA) models using 593 data points and smoothened case and covariate time-series data to generate a 28-day forecast of COVID-19 case trends during the third wave in Malaysia. SARIMA models were developed using COVID-19 case data sourced from the Ministry of Health Malaysia’s official website. Model training and validation was conducted from 22 January 2020 to 5 September 2021 using daily COVID-19 case data. The SARIMA model with the lowest root mean square error (RMSE), mean absolute percentage error (MAE) and Bayesian information criterion (BIC) was selected to generate forecasts from 6 September to 3 October 2021. The best SARIMA model with a RMSE = 73.374, MAE = 39.716 and BIC = 8.656 showed a downward trend of COVID-19 cases during the forecast period, wherein the observed daily cases were within the forecast range. The majority (89%) of the difference between the forecasted and observed values was well within a deviation range of 25%. Based on this work, we conclude that SARIMA models developed in this paper using 593 data points and smoothened data and sensitive covariates can generate accurate forecast of COVID-19 case trends.

## 1. Introduction

The novel coronavirus, also known as COVID-19, was first detected in Wuhan, China on 31 December 2019. Within a short period, this virus has spread globally affecting many countries worldwide [1,2]. As a result, the World Health Organization (WHO) declared COVID-19 a Public Health Emergency of International Concern (PHEIC) on 30 January 2020, and subsequently declared it a pandemic on 11 March 2020 [2]. Since then, many countries globally have implemented several non-pharmaceutical interventions (NPIs) to contain the spread of COVID-19 and control the outbreak. However, despite these measures, there have been reports of resurgence of cases resulting in recurrent waves of COVID-19 infection worldwide [3,4,5,6].

Similarly, Malaysia reported its first wave of COVID-19 infection from 25 January to 16 February 2020 involving 22 cases, of which the majority were imported cases from China. Following this, the second wave occurred from 27 February to 19 September 2020 involving 10,167 cases. Currently, Malaysia is experiencing a resurgence with the third wave, which began on 20 September 2020 [7,8]. As of 3 October 2021, Malaysia has reported 2,277,565 (7% per total population) positive cases with 26,212 deaths, with a 7-day incidence rate of 34.9 per 100,000 populations [9]. In view of COVID-19 being a novel disease with limited baseline data and uncertain and exponential case trends, forecasting this disease will assist health authorities in planning and implementing outbreak control strategies efficiently and effectively. Hence, the modeling of COVID-19 case trends has been widely explored and adopted as an approach to assist in controlling infectious disease outbreaks [10,11].

Several studies have modeled COVID-19 case trends to guide decision making on the management and control of the pandemic [12]. In addition, forecasting case trends has been able to function as an early warning system for authorities to react by instituting appropriate NPI’s such as the movement control order (MCO), use of face masks, hand hygiene practice, and physical distancing to prevent disease outbreaks [13,14]. Compartmental models (e.g., the SEIR or Susceptible-Exposed-Infected-Removed and the SIR or Susceptible-Infected-Removed epidemic models) have been used to forecast COVID-19 case trends globally; however, these models require complex analysis, can be time consuming to develop, and require a wide range of parameters on the disease transmission dynamics for the model development [15,16,17,18].

To address these challenges, where disease parameters are not available, especially for novel diseases like COVID-19, time-series models such as the Auto-Regressive Integrated Moving Average (ARIMA) or Seasonal ARIMA (SARIMA) models can be used as suitable modeling methods that are relatively simple to develop and are able to produce forecasts with a single time-series dataset [19]. ARIMA is a prediction model used for time-series analysis and forecasting based on temporally evolving data where the observations of well-defined attributes are obtained through repeated measurements over time [20]. Whereas SARIMA specifically describe the seasonal models. Due to this, ARIMA models have been frequently used for modeling infectious diseases such as Hepatitis A, Severe Acute Respiratory Syndrome (SARS) and Dengue fever [21,22]. In addition, several recent studies have developed ARIMA and SARIMA models to forecast COVID-19 case trends in India, Sweden, Italy, Spain, France and Saudi Arabia [19,23,24,25,26,27], wherein these studies have shown that by using ARIMA and SARIMA models, forecasts of novel diseases such as COVID-19 can be generated accurately [23,24,25,26,27].

In Malaysia, ARIMA models were developed to forecast daily COVID-19 case trends during the second wave from 18 April to 1 May 2020 [28]. Being a novel disease, the data points available to develop ARIMA models were limited. However, limited time-series data were able to forecast the COVID-19 case trends reasonably well using ARIMA models [28]. Evidence in literature suggests that a minimum of 50 data points is required for developing ARIMA models. However, it is universally agreed that more data points would generally improve the model accuracy [29]. Previous work on ARIMA modeling in Malaysia was able to generate a 14-day forecast during the first phase of movement control order (MCO) [28]. To expand from this previous work, the availability of more data points would be important in improving the ARIMA model forecast accuracy.

While time-series models are simple to develop and can generate accurate forecasts, few studies to date have forecasted COVID-19 case trends using these models in Malaysia [28,30]. Therefore, this study aims to develop SARIMA models using more data points with appropriate covariates and smoothening strategies to provide a 28-day forecast during the third wave of COVID-19 in Malaysia.

## 2. Materials and Methods

### 2.1. Data Source

National- and state-level COVID-19 daily cases from 22 January 2020 to 5 September 2021 (593 data points) were sourced from the Ministry of Health (MOH) Malaysia official website (http://covid-19.moh.gov.my/ (accessed on 1 November 2021)) [31].

### 2.2. Data Analysis

Statistical Package for the Social Sciences (SPSS) version 26.0 (IBM Corp. Released 2019. IBM SPSS Statistics for Windows, Version 26.0. Armonk, NY, USA: IBM Corp) was used to develop, train and validate the time-series (SARIMA and ARIMA) models. Time-series models with a Ljung Box test (18) with a non-significant p-value, the lowest Bayesian information criterion (BIC), the lowest root mean square error (RMSE) and lowest mean absolute error (MAE) were selected to generate forecasts of COVID-19 cases for a duration of 28 days (6 September 2021 to 3 October 2021). The Augmented Dicky Fuller (ADF) test was performed in R programming software version 4.0.3 to confirm data stationarity, wherein *p* values < 0.05 indicate that the data are stationary [32]. In addition, the accuracy of the 28-day forecast was analyzed by calculating the deviation index based on the formula below:(1)$$\mathrm{Accuracy}=\sum^{\;}\frac{\left(7-\mathrm{day}\;\mathrm{MA}\;\mathrm{of}\;\mathrm{daily}\;\mathrm{observed}\;\mathrm{cases}\;\mathrm{on}\;\mathrm{day}\;t\;-\;\mathrm{Model}\;\mathrm{forecast}\;\mathrm{on}\;\mathrm{day}\;t\right)}{\mathrm{Model}\;\mathrm{forecast}\;\mathrm{on}\;\mathrm{day}\;t\;}\times 100\%$$
where the summation extends over all 28 days from the beginning until the end of the validation period, and the 7-day moving average (MA) stands for the 7-day moving average of daily cases based on a window centered around each day t. Prior to analysis, the case data were validated with other data sets, such as COVID-19 daily and ICU admission, and deaths by performing a correlation and cross-correlation analysis. The case data were significantly correlated with COVID-19 daily admission, ICU admission and deaths with the Spearman’s correlation coefficient ranging from 0.920 to 0.945. The cross-correlation findings indicated that positive significant correlations were present at lag 0, which remained significant with subsequent lags. This justifies the validity of the case data used in this study.

### 2.3. Model Development

The model development of SARIMA consists of two stages, namely, parameters estimation and model identification. The first stage was to estimate the non-seasonal (p,d,q) and seasonal (P,D,Q) parameters, where p refers to the order of autoregressive (AR) component, d refers to the degree of differencing of the original time series and q refers to the order of the MA component. These were determined by the autocorrelation function (ACF) and partial autocorrelation function (PACF). In the parameter estimation stage, the unknown coefficients corresponding to the AR and MA components of the ARIMA model were estimated by implementing a multivariate regression analysis [23]. All data were transformed into a time-series format following which, due to high degree of data variability, seven-day moving averages were estimated. This assisted the identification of true case trends and removed noise from the data set. Subsequently, the data were checked and confirmed for stationarity using the ADF test. The ACF and PACF residuals were visually examined to identify the existence of seasonality of the time-series data, wherein a significant spike was observed at lag 7 suggesting a 7-day seasonal AR component.

In the model identification stage, independent and dependent variables were used in several combinations in order to identify the best SARIMA model. The independent variable (covariate) used in the model identification was the state-level COVID-19 daily case data. We used case data from the state of Selangor as the covariate. The covariate was found to have no evidence of multicollinearity by using the variance inflation factor (VIF) test [33]. The dependent variable used was the 7-day moving average for the national daily COVID-19 cases. A 7-day moving average was used because the daily COVID-19 cases were found to show a high degree of variability (noise) and skewness, determined by estimating the variance and the test of normality, respectively. To reduce noise and skewness, the data were averaged on a moving window using a 7-day moving average smoother (Figure A1) [28]. Following using a 7-day moving average, the variance reduced from 28,627,924 to 26,952,897. Subsequently, three models were identified, and the best model was selected based on Ljung Box test with a non-significant *p*-value, the lowest RMSE, lowest BIC and lowest MAE.

### 2.4. Model Training, Validation and Forecast Generation

A total of 415 data points were used for model training (22 January 2020 to 11 March 2021) and 178 data points were used for validation (12 March 2021 to 5 September 2021). This corresponds to a 70–30% partition of total data points for model training and validation, which is suggested as the optimal ratio in the literature [19]. During the model training and validation periods, the model fit was ascertained by comparing the estimated case numbers with observed COVID-19 case numbers as well as ensuring observed cases are within the 95% confidence interval (CI) of the model forecasts. The best fitting SARIMA model was selected based on the lowest MAE, RMSE and BIC values, which were then used to generate a 28-day forecast (6 September 2021 to 3 October 2021) of daily COVID-19 case numbers. The forecasted case numbers were then compared to the observed case numbers during the forecast period visually as well as by estimating the deviation index.

## 3. Results

This study developed three different models arising from combinations of (p,d,q) (P,D,Q) with the inclusion or exclusion of seasonal component and the independent covariate. The three models used are shown in Table 1. The SARIMA model (1,2,1) (2,0,0) with covariates of daily COVID-19 cases in Selangor (7-day moving average) was selected as the best fit model with MAE = 39.716, BIC = 8.656 and RMSE = 73.374. The generated ACF and PACF plots for the original time-series ARIMA model show that the correlations gradually trail off as a function of the lag, thus indicating a non-stationary series as shown in (Figure A2). Due to the non-stationary time-series data (ADF test statistic = −0.84, *p* value > 0.05), a two-degree differencing was applied to transform the series to stationarity as shown in Figure 1. Subsequently, the ADF test statistic was applied to the differenced time series and yielded a value of −6.216 with *p*-value < 0.05, therefore confirming stationarity. In addition, the model was validated using the Ljung-Box Q test, which suggested that the ACF for the residuals at different lag times was not statistically different from zero (the Ljung-Box Q test statistic (18) value was 52.628). The majority of the ACF and PACF residuals were within the 95% CI as shown in Figure 2. Table 2 shows the estimated model parameters for SARIMA (1,2,1) (2,0,0) based on the training data.

The SARIMA model (1,2,1) (2,0,0) forecasts showed a downward trend of COVID-19 cases during the forecast period from 6 September 2021 to 3 October 2021 as shown in Figure 3. The observed cases during the forecast period also showed a similar downward trend where the observed daily cases (7-day MA) were within the 95% CI of the model forecast as shown in Figure 4. Daily forecasted values and its 95% CI are shown in Table A1). The difference between the forecasted and observed values was well within a negative deviation range of 25%, wherein a total of 25 observations out of the 28 data points (89%) were within this deviation range as shown in Figure 5.

## 4. Discussion

Being a novel disease, a detailed understanding of COVID-19 disease dynamics is limited resulting in difficulties to predict disease outbreaks. However, to ensure a balance between lives and livelihood, forecasting of disease outbreaks has become an important approach in assisting the management and control of the COVID-19 pandemic. It is evidence in the literature that forecasting using ARIMA models was able to provide reliable forecast results during the second wave of COVID-19 pandemic in Malaysia [28].

Despite having limited and stochastic data, a 7-day cyclical trend was observed in the time-series data, whereby a reduction of cases was usually observed during the weekends, which could be due to variation in data collection and surveillance during weekends. Hence, analysis using a seasonal component was carried out to determine the most appropriate time-series model. This is the first study to develop SARIMA models to forecast COVID-19 cases in Malaysia during the third wave (6 September 2021 to 3 October 2021), wherein this study found that SARIMA model (1,2,1) (2,0,0) was able to provide an accurate forecast despite the presence of limited stochastic data. To improve forecasts using SARIMA models, several measures were taken. Among them, these include using 593 data points for model development, smoothing time-series data of both the dependent and independent variables, which improved the identification of signals, and using appropriate sensitive covariates.

The study found that with the available number of data points (*n* = 593), SARIMA models could be developed to generate forecasts of COVID-19 case trends up to a duration of 28 days. This is evident by the accurate forecasts of the COVID-19 case trends generated by the SARIMA model (1,2,1) (2,0,0) which were similar to the observed downward case trends during the forecast period. In addition, the observed cases (7-day MA) were within the 95% CI of the model forecasts. Furthermore, 89% of the differences between the forecasted and observed values were well within a deviation range of 25%. Therefore, these findings indicate that the SARIMA model (1,2,1) (2,0,0) generated an accurate forecast. Similar findings have been reported in the literature, wherein 593 data points generally improved the SARIMA model accuracy [29]. Therefore, continuous efforts to develop SARIMA models to forecast COVID-19 case trends should be attempted by using more data points as time progresses.

The COVID-19 outbreak is dynamic in nature and rapidly changes resulting in fluctuating daily case numbers. As a result of this, the COVID-19 time-series data are inherently noisy with obscure trends, which make the identification of true data trends and signals for the development of SARIMA model challenging. In order to reduce the noise in the time-series data, a 7-day moving average for the dependent variable (daily COVID-19 national cases) was used so that the model may transmute the true data trends into signals. This, in turn, improved the overall trend of the dependent variable and provided better signals for data training and validation, which subsequently resulted in a more accurate forecast. Similar approaches using moving averages have been utilized in the past to obtain true signals from time-series data when developing ARIMA models [28].

In addition, to improve the SARIMA model forecast, appropriate covariates such as the daily subnational Selangor state case report was used, as it accounts for the majority of the COVID-19 cases and has been consistently reporting high numbers of COVID-19 cases since the beginning of the pandemic in Malaysia. Furthermore, the case trends in Selangor state closely mirrored the observed national daily COVID-19 case trends. To further improve the model, as with the national’s time-series data, the COVID-19 daily cases in Selangor were also smoothened using a 7-day moving average in order to reduce the noise and improve signal detection. The generated goodness of fit showed very minimal difference between the smoothened and non-smoothened SARIMA model, but in terms of the model forecast, the model forecast with smoothened covariates showed a better resemblance to the actual COVID-19 daily cases. This study found that using a single appropriate covariate (Selangor state COVID-19 incidence with smoothening) is sufficient to develop accurate SARIMA models. This could be due to the selection of covariates that are truly representative of the COVID-19 case trends as well as having the ability to reflect the national-level trends to further improve the model forecast.

Daily data were used in this study, and the weekly variation on reporting resulted in the seasonal model pattern observed in the study. Previous papers published by Hernandex et al. (2020) and Alraj et al. (2021) report ARIMA (2,1,5) and ARIMA (2,0,2) for forecasting COVID-19 cases using time-series model, respectively, which are different from the SARIMA model reported in this study. This variation could be attributed to methodological variations such the use of data smothering, covariates and seasonal effects as applied in this study, which were not reported in the work done by Hernandex et al. (2020) and Alraj et al. (2021) [34,35]. In addition, the SARIMA model developed in this study was not based on polynomial functions or hybrid compartmental models, as done by Hernandex et al. (2020) and Alraj et al. (2021) [34,35]. The SARIMA model developed in this study reports the model goodness-of-fit estimates (i.e., RMSE = 73.37 and MAE = 39.72), which are better compared to previous ARIMA models developed by Hernandex et al. (2020) (i.e., RMSE = 89.46) and Tabar et al. (2021) (i.e., RMSE = 259.98 and MAE = 144.84) [34,36]. In this study, the observed cases were well within the forecast range of model, with majority of the deviation index being within a range of 25%. Similar findings were reported by Hernandex et al. (2020) and Alraj et al. (2021), wherein the time-series models provided accurate forecast trajectories with fairly narrow deviation index [34,35].

There are several strengths of this study. First, a total of 593 data points were used for model training and validation. Second, data were smoothened to remove noise to obtain better signals. Additionally, we included a smoothened covariate that was able to generate good signals for the model development. Overall, these measures resulted in the development of an appropriate SARIMA model, which was able to produce reasonably accurate forecasts of COVID-19 cases. The duration of the data used for model development remain the main limitation in this study. However, this limitation was addressed by using covariates to develop the model. Future studies should look into developing time-series models with the availability of more data points as time progresses as well as using more covariates that are reflective of the transmission dynamics.

Since COVID-19 is a novel disease with continuously changing disease dynamics, using time-series data (*n* = 593) and appropriate covariates would provide more signals to train and validate the models, hence resulting in more accurate forecasts. This is especially important as the third wave has been by far the most challenging one faced by the health authorities in Malaysia. In addition, this study uses smoothened data to obtain more appropriate signals during the model development in order to improve the model accuracy.

## 5. Conclusions

This study showed that SARIMA models developed using 593 data points with sensitive covariates and smoothened data can generate a reliable and accurate forecast of COVID-19 case trends up to 28 days in Malaysia. The SARIMA models that were developed were used to advise the extension of the national MCO from phase 1 to 2 during the early stages of outbreak in Malaysia [28]. Modeling the COVID-19 case trends during the pandemic has been essential in assisting authorities on deciding the appropriate time to implement, relax, lift, strengthen or reinstitute public health social measures such as the MCO and standard operating procedure (SOP) in managing the COVID-19 outbreak in Malaysia.

## Figures and Tables

**Figure 1 ijerph-19-01504-f001:**
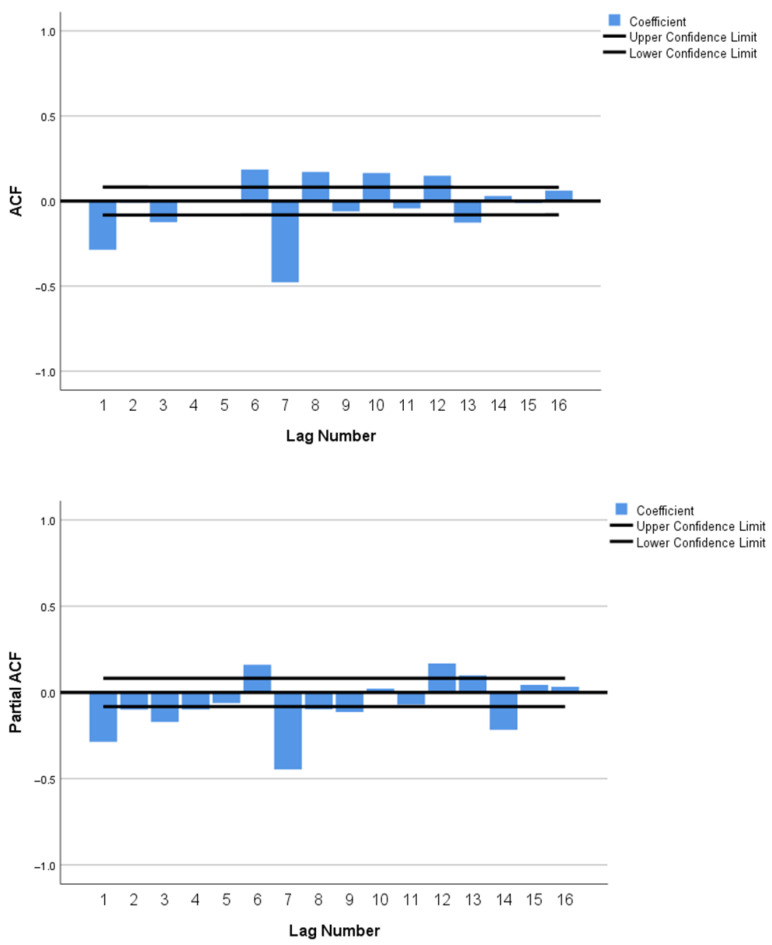
ACF plots and PACF plots with two-degree differencing.

**Figure 2 ijerph-19-01504-f002:**
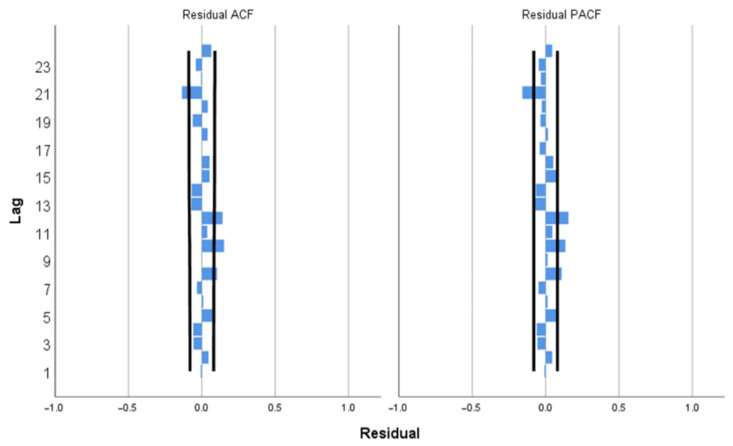
ACF and PACF residuals for SARIMA (1,2,1) (2,0,0).

**Figure 3 ijerph-19-01504-f003:**
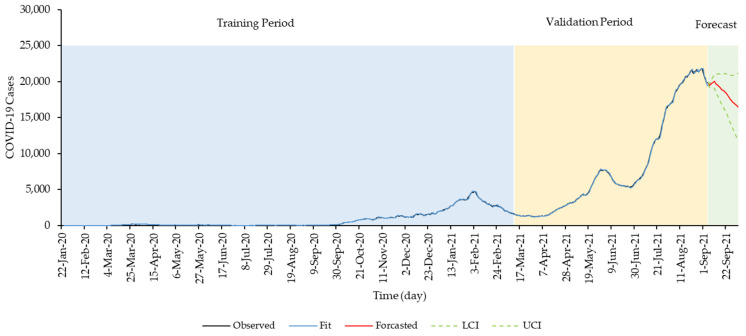
SARIMA (1,2,1) (2,0,0) model validation and forecast from 22 January 2020 to 3 October 2021, Malaysia (28-day forecast).

**Figure 4 ijerph-19-01504-f004:**
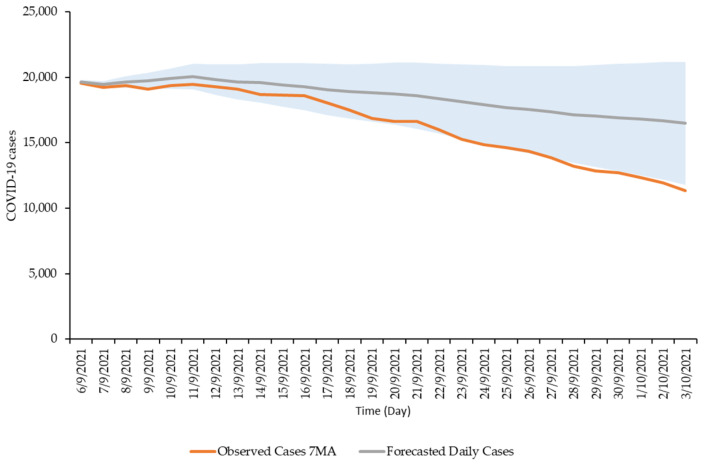
SARIMA (1,2,1) (2,0,0) model forecasted and the 7-day MA observed daily COVID-19 cases from 6 September 2021 to 3 October 2021, Malaysia (28-day forecast).

**Figure 5 ijerph-19-01504-f005:**
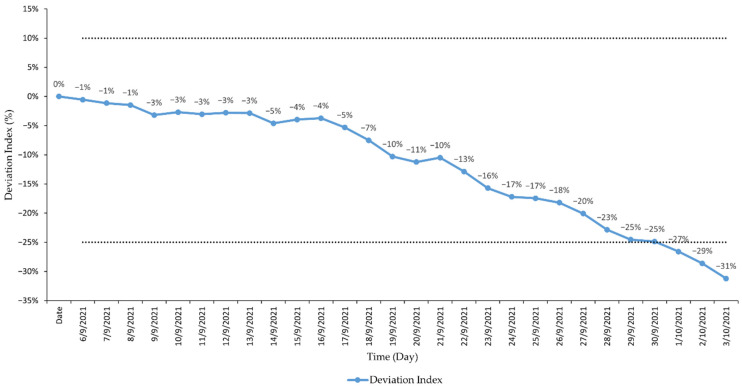
Deviation index of the SARIMA (1,2,1) (2,0,0) forecast from 6 September 2021 to 3 October 2021, Malaysia.

**Table 1 ijerph-19-01504-t001:** ARIMA models.

Model	RMSE	MAE	BIC	Q18
SARIMA (1,2,1) (2,0,0) with covariates—Selangor daily cases *	73.374	39.716	8.656	52.628
SARIMA (1,2,1) (2,0,0) **	73.704	39.111	8.654	51.505
SARIMA (1,2,1) (2,0,0) with covariates—Selangor daily cases **	73.584	39.436	8.662	51.812
SARIMA (1,2,1) (2,0,0)	524.894	277.137	12.580	68.129
ARIMA (1,2,1) with covariates—Selangor daily cases *	87.301	47.836	8.982	167.963

Note: * smoothened dependent and independent variables using 7-day moving average; ** smoothened dependent using 7-day moving average.

**Table 2 ijerph-19-01504-t002:** Model parameters for SARIMA (1,2,1) (2,0,0).

Model Covariate		Estimates	Standard Error (SE)	*p*-Value *
Malaysia (7-day MA)	Constant	1.157	0.988	0.242
	AR (Lag 1)	0.251	0.116	0.031 *
	Difference	2		
	MA (Lag 1)	0.564	0.098	0.000 *
	AR, Seasonal (Lag 1)	−0.710	0.044	0.000 *
	AR, Seasonal (Lag 2)	−0.365	0.047	0.000 *
Daily Selangor cases (7-day MA)	Numerator (Lag 0)	−0.001	0.000 **	0.011 *

* *p*-value < 0.05 indicates significance for the estimated coefficients in the fitted ARIMA model (1,2,1) (2,0,0), ** actual value: 0.000493.

## Data Availability

Sourced from the Ministry of Health (MOH) Malaysia official website (http://covid-19.moh.gov.my, accessed on 1 November 2021).

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
