# Peer review of "Forecasting COVID-19 Case Trends Using SARIMA Models during the Third Wave of COVID-19 in Malaysia"

_ijerph, 2022, doi:10.3390/ijerph19031504_

Round 1

Reviewer 1 Report

The paper presents a forecast of Covid using ARIMA models, Despite the good intentions of the authors bt the paper needs more work. I can not see the novelty in this work, I have the following suggestions to improve the paper:

  • ARIMA must be explicated a detail.
  • Comparison with other approaches must be presented.
  • In general, the paper needs more work.

Reviewer 2 Report

The paper is a case study on modelling and forecasting Malaysia COVID cases. Authors use ARIMA models which is a quite common tool for time series. My comments are listed below for the method and data analysis. 

  1. The authors claim in the introduction that there are many modelling tools for COVID case forecasting but they are too complicated. It may be true that ARIMA models are much simpler to use with ready-to-use software packages, it cannot model the physical dynamics for COVID spreading, unlike something like SIR models. It may forecast the trend (up or down) satisfactorily, I don't think the model has an ability to forecast the peak - due to that the model cannot incorporate the underlying physical driver of the time series.
  2. Related to Point 1, there are many alternative methods for modeling the COVID trend. Then why don't the authors compare their model performance with any other existing methods? It is unusual that there is no such comparison. 
  3. The authors selected candidate models ARIMA (1,1,7) and (1,1,8). Normally we wouldn't have such a high order for p,d,q - even in some very complicated financial time series. I doubt q=7 or 8 for the MA component is overparameterized. In this case, you will need to check for redundancy (are all parameters significant?) (Refer to the textbook Time Series Analysis by Cryer and Chan, Chapter 8). 
  4. The authors choose p-value<0.05 for Ljung-Box test. Is this a mistake? Ljung-Box test is for testing independence of residuals. The null hypothesis is that residuals are independent. P-value<0.05 is a rejection to the null indicating residuals are not independent -- meaning your modelling is not adequate. I don't know whether there is a statistician on board - but you certainly need one for such a pure application of statistics. 

Reviewer 3 Report

The work "Forecasting COVID-19 Case Trends Using ARIMA Models During the Third Wave of COVID-19 in Malaysia" by Cia Vei Tan et al. employ the Auto-Regressive Integrated Moving Average single time-series analysis for forcasting the COVID-19 Case trends.

Although the work has potential interest, I think it is in the current form not appropriate for publication in the International Journal of Enviromental Research and Public Health.

Some important issues should to be addressed and explained in the paper 

1) In the work long data-sets are used, spanning more than a year of the epidemics. In this time, the dynamics of the COVID-19 disease has been relatively well studied. Therefore epidemiological models that have this knowledge incorporated and can be still easy to run, should probably give more accurate forecasts (for example, compartmental SEIR-like models). It should therefore be clearly explained in the paper why and when the presented model, which does not include the mechanisms of the epidemiological dynamics, should be used.

2) A time-series of COVID-19 daily cases in Malaysia is used. However, the number of daily cases depends on the number and types of tests performed and on the national strategy of testing. This has probably changed during the epidemics. Therefore at least a validation with a different sets of data should be considered (for example the number of hospitalizations, Intensive Care Unit cases and/or deaths due to COVID-19).

3) In Conclusions it is stated that modelling the COVID-19 case trends has been essential in assisting the authorities. If this refers to the here presented model and analysis, it would be very interesting for the readers to know more of how, when and in what form it was used by the decision makers.

4) Precise scientific expressions should be used throughout the paper. For example, expressions "longer data" (in Abstract) and  "longer data point" (in Conclusions) are not well defined.

Other minor comments:

  • Ref. 11 - journal/source is missing
  • for better illustration, also proportions should be given together with the absolute values of COVID-19 cases in Malaysia in the 2nd paragraph of Introduction (for example 7 or 14-days incidence per 100 000 people).

Round 2

Reviewer 1 Report

The results must be compared with other papers, I suggest some papers to compare

https://www.sciencedirect.com/science/article/pii/S1568494620305482

https://www.sciencedirect.com/science/article/pii/S156849462030870X

https://www.sciencedirect.com/science/article/pii/S2468042720301032

Reviewer 2 Report

Observing Figure 1 and Figure 2, clearly there is a significant spike at lag 7, which implies the model with AR(1) component is inadequate. Given the experience of COVID case reporting and observing the raw data plot in the appendix, this lag 7 autocorrelation is likely a seasonal AR component (7 days a week). Therefore, I think the appropriate model (after differencing) should contain AR(1) and a seasonal AR(1). Authors can refer to Cryer and Chan Chapter 10. 

Minor: Table 2 last column Significance >0.05: what do you mean here? Significance for what? 

Reviewer 3 Report

I believed the revisions made to the manuscript provided enough clarity and additional information for this work 

Round 3

Reviewer 2 Report

This version has addressed my concerns.